# LongVILA: Scaling Long-Context Visual Language Models for Long Videos

Yukang Chen[1][*]    Fuzhao Xue[1][*]    Dacheng Li[3][†]    Qinghao Hu[2][†]

Ligeng Zhu[1]    Xiuyu Li[3]    Yunhao Fang[1]    Haotian Tang[1,2]    Shang Yang[2]

Zhijian Liu[1]    Ethan He[1]    Hongxu Yin[1]    Pavlo Molchanov[1]    Jan Kautz[1]

Linxi Fan[1]    Yuke Zhu[1,4]    Yao Lu[1]    Song Han[1,2]

[1]NVIDIA    [2]MIT    [3]UC Berkeley    [4]UT Austin

## Abstract

Long-context capability is critical for multi-modal foundation models, especially for long video understanding. We introduce LongVILA, a full-stack solution for long-context visual-language models by co-designing the algorithm and system. For model training, we upgrade existing VLMs to support long video understanding by incorporating two additional stages, *i.e.*, long context extension and long video supervised fine-tuning. However, training on long video is computationally and memory intensive. We introduce the long-context Multi-Modal Sequence Parallelism (MM-SP) system that efficiently parallelizes long video training and inference, enabling 2M context length training on 256 GPUs without any gradient checkpointing. LongVILA efficiently extends the number of video frames of VILA from 8 to 2048, achieving 99.8% accuracy in 6,000-frame (more than 1 million tokens) video needle-in-a-haystack. LongVILA-7B demonstrates strong accuracy on 9 popular video benchmarks, *e.g.,* 65.1% VideoMME with subtitle. Besides, MM-SP is $2.1\times$ - $5.7\times$ faster than ring style sequence parallelism and $1.1\times$ - $1.4\times$ faster than Megatron with a hybrid context and tensor parallelism. Moreover, it seamlessly integrates with Hugging Face Transformers. Our code and models are available at github.com/NVlabs/VILA/longvila.

## 1 Introduction

Integrating multi-modal understanding with long-context capability is important. A foundation model supporting more modalities can take more flexible input signals so that people can interact with the model in more diverse manners, *e.g.,* GPT-4o-like multi-modal chatbot, multi-modal web agent (Koh et al., 2024), and real-world robotics foundation model (Brohan et al., 2022; 2023; Padalkar et al., 2023). Longer context enables models to process more information, *e.g.,* long documents, repo-level codebase, and hour-length video, which similarly provides required features to more real-world applications.

While some works have enabled long-context Vision-Language Models (VLMs) (Lin et al., 2023b; Weng et al., 2024), they employ simplified approaches rather than offering a comprehensive solution. For instance, LongVA (Zhang et al., 2024b) relies on long-context LLMs and trains models on short-context data. LongVLM (Weng et al., 2024) utilizes token compression to circumvent context extension. These approaches sidestep more challenging issues, such as the development of a robust long-context multi-modal training framework and corresponding dataset design.

A full-stack design is crucial for long-context Vision-Language Models (VLMs). Training large models is typically a complex, systematic endeavor that demands both data engineering (Betker et al., 2023; Ouyang et al., 2022; Zhou et al., 2024) and system-software co-design (Lepikhin et al., 2020; Chowdhery et al., 2023; Shoeybi et al., 2019; Brown et al., 2020; Dehghani et al., 2023). Unlike text-only LLMs, VLMs (*e.g.,* LLaVA (Liu et al., 2023c)) often require distinct model architectures and flexible distributed training strategies. Additionally, long-context modeling necessitates

---

[*]Algorithm Lead. [†] System Lead. The first four authors have equal contributions.

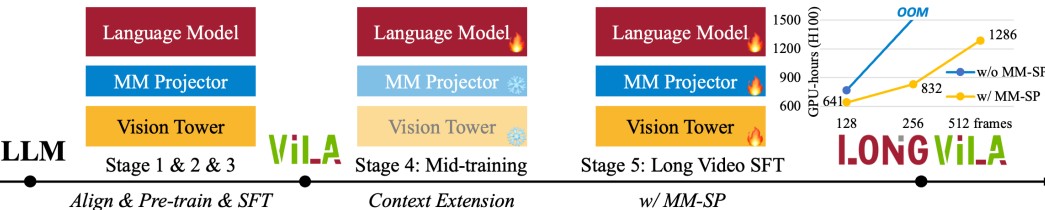

Figure 1: The LongVILA training pipeline. In Stages 1 through 3, the process starts with alignment, pre-training, and supervised fine-tuning. In Stage 4, the model undergoes mid-training context extension. Finally, in Stage 5, the model is fine-tuned for long video understanding with Multi-Modal Sequence Parallelism (MM-SP).

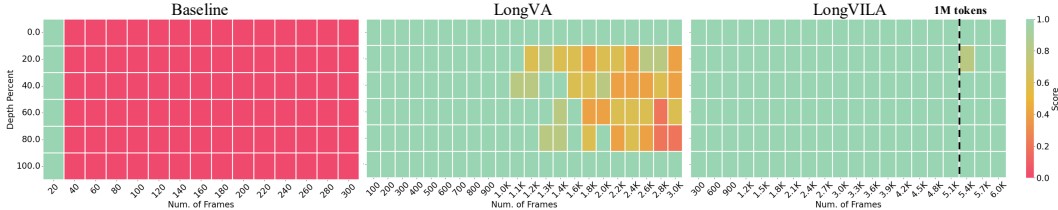

Figure 2: Comparison of Needle in the Long Video Haystack Experiment. The 32-frame baseline model (left) can not retrieve right needles after 32 frames. LongVA (middle) achieves 87.6% accuracy in 3,000 frames. In contrast, the LongVILA model (right), trained on 2048 frames, presents 99.8% accuracy on 6,000 frames (more than 1 million context length).

not only long-context data to fully utilize the model's capabilities (Fu et al., 2024c; Chen et al., 2023) but also infrastructure capable of supporting memory-intensive long-context training (Li et al., 2021; Jacobs et al., 2023; Li et al., 2023a). Therefore, a full-stack design, encompassing training pipeline and system, is indispensable for long-context VLMs.

In this work, we introduce LongVILA, a comprehensive solution for long-context VLMs. For training **pipeline**, we implement a five-stage training curriculum as Figure 1: (1) multi-modal alignment, (2) large-scale pre-training, (3) short supervised fine-tuning, (4) context extension for LLMs, and (5) long supervised fine-tuning. For **system**, we establish an efficient and user-friendly framework, namely Multi-Modal Sequence Parallelism (MM-SP), which supports training and inferencing memory-intensive long-context VLMs.

LongVILA-7B presents strong performance on 9 popular benchmarks, *e.g.,* 65.1% on VideoMME (Fu et al., 2024a) with subtitle. The LongVILA model, trained on 2048 frames, achieves 99.8% accuracy in the needle-in-a-haystack experiments with 6,000 frames, with a context length of more than 1 million tokens. In ablations, by increasing the number of video frames using LongVILA, the performance on VideoMME in long videos consistently improves (Figure 3). Our MM-SP system can efficiently scale the context length up to 2 million tokens without gradient checkpointing, achieving 2.1× to 5.7× speedup compared to ring style sequence parallelism, and 1.1× to 1.4× compared to Megatron with a hybrid context parallelism and tensor parallelism.

## 2 RELATED WORKS

**Visual language model architecture.** There are two predominant designs for VLMs: the encoder-decoder architecture (*e.g.,*, LLaVA (Liu et al., 2023c), PaLM-E (Driess et al., 2023)) and the decoder-only architecture (*e.g.,*, Fuyu (Bavishi et al., 2023), Chameleon (Team, 2024)). Encoder-Decoder VLMs connect the vision encoder to the LLM decoder through a multi-modal projector. Certain multi-modal projectors, such as spatial pooling and Q-former, significantly reduce the number of tokens per image or video frame, thereby lowering the computational burden on the LLM decoder. In contrast, decoder-only LLMs typically process raw patches as input without hierarchical token pooling, making it more challenging to reduce the token count for each image or frame. In this work, we build on VILA (Lin et al., 2023b) as our foundation. It is worth noting that enhanced variants

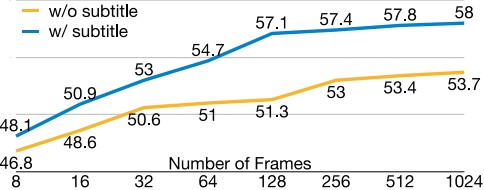

| Training | VideoMME | | | |
|---|---|---|---|---|
| Stages | Average | Short | Medium | Long |
| 1-2-3-4-5 | 57.5 | 69.3 | 56.1 | 47.0 |
| 1-2-4-(3&5) | 55.9 | 67.4 | 54.1 | 46.1 |
| 4-1-2-3-5 | 56.0 | 69.2 | 54.1 | 44.5 |
| 4-1-2-(3&5) | 55.3 | 67.2 | 53.6 | 45.1 |

Table 1: Ablations on various training stage settings on VideoMME (without subtitle). (3&5) means joint training of the stage 3 and 5.

Figure 3: Scaling video frames improves VideoMME accuracy in long category.

of VILA exist, such as VILA[2](Fang et al., 2024) for improved performance and X-VILA(Ye et al., 2024) for cross-modality understanding, reasoning, and generation. For our model architecture and training pipeline, we adhere to the standard VILA-1.5 version.

**Sequence parallelism and hybrid strategy.** Long-context training examples often exceed the memory capacity of a single device. To address this issue, the sequence parallelism paradigm has been widely adopted in the text-only LLM community, distributing a single sequence across multiple devices. Specifically, Ring-style systems Li et al. (2021; 2023a); Liu et al. (2023a) use Point-to-Point (P2P) communication primitives to collectively compute the attention module, while DeepSpeed-Ulysses Jacobs et al. (2023) employs an All-to-All (A2A) primitive to alternate between sharding the sequence dimension and the attention head dimension during attention computation. Ulysses generally achieves higher throughput than Ring-style SP due to its more efficient A2A communication primitive and larger, unsegmented computation blocks. However, its scalability is limited by the number of attention heads. Recently, USP (Fang & Zhao, 2024) was introduced as the first to integrate Ring-style SP and Ulysses SP, combining the strengths of both approaches. LoongTrain (Gu et al., 2024) further optimizes communication and placement strategies to enhance training efficiency. Following (Fang & Zhao, 2024; Gu et al., 2024), we extend the system to multi-modal scenarios to accommodate complex attention masks and variable-length input sequences. Our work is the first to design and implement a sequence parallelism system for visual language models.

# 3 LONGVILA TRAINING PIPELINE

As shown in Figure 1, in our pipeline, there are five training stages, *i.e.*, Stage 1: multi-modal alignment, Stage 2: large-scale pre-training, Stage 3: supervised fine-tuning, Stage 4: context extension for LLM, Stage 5: long supervised fine-tuning. Stage 1, 2, and 3 follow VILA (Lin et al., 2023b), to firstly bridge the gap between LLM and vision encoder, and then pre-training on larger datasets. In Stage 1, only the multi-modal projector is trainable with others frozen. In Stage 2, we freeze the vision encoder and training LLM and the multi-modal projector. In Stage 3, we fully fine-tune the model for short data instruction following, *e.g.*, image and short video datasets. Afterwards, we extend the context length of LLM with text-only dataset in a continued pre-training manner in Stage 4. In Stage 5, we adopt our MM-SP system (§4) to enhance the instruction following abilities by long video supervised fine-tuning. It is noted that all parameters are trainable in the final stage.

## 3.1 STAGE1&2&3: ALIGNMENT, PRE-TRAINING, AND SHORT SUPERVISED FINE-TUNING

We first use open-sourced image and video caption datasets to train the multi-modal projector in stage (1) to conduct the multi-modal alignment. Note that, following (Lin et al., 2023b), both vision encoder and LLM decoder are frozen at this stage. After that, we conduct large-scale pre-training to learn general multi-modal capacity at scale. To improve the quality of large open-sourced datasets, we follow VILA[2] (Fang et al., 2024) to relabel COYO-25M (Lin et al., 2023b; Byeon et al., 2022) with VILA-1.5-40B (Lin et al., 2023b). The supervised fine-tuning process incorporates mixed data types, including both images and videos. For short video comprehension, we utilize open-source video instruction-following datasets, *e.g.,* YouCook2 Zhou et al. (2018) and ShareGPTVideo Zhang et al. (2024c). In experiments, our model is based on Qwen2-1.5B and Qwen2-7B (qwe, 2024).

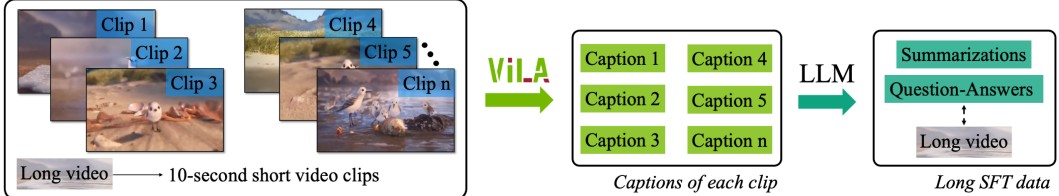

Figure 4: The proportion of question and video categories in our LongVILA_sft dataset. We have 15,292 videos in total. For each video, there are one sample for captioning and the other question.

Figure 5: The pipeline for generating instruction-following data from long videos. The process begins by segmenting the long video into short clips, each approximately 10 seconds in length. These clips are individually annotated with captions using the VILA-1.5 model. Subsequently, an LLM is employed to generate question-and-answer pairs based on the captions of these clips. Generated questions include summarization and other inquiries pertinent to the content of long videos.

## 3.2    STAGE4: CONTEXT EXTENSION FOR LLMS

Our empirical research indicates that extending the context length of LLMs is essential prior to engaging in supervised fine-tuning with long video datasets. Following Stage 2 of our methodology, we execute a continuation of pre-training on the LLM to enhance its context length to 262,144, utilizing a total of 17B tokens. We employ a progressive training schedule, incrementally increasing the context length from 8,192 to 65,536, and ultimately to 262,144, utilizing the SlimPajama dataset (Soboleva et al., 2023) in accordance with the methodology outlined by (Fu et al., 2024d).

Furthermore, we augment the base frequency of the Rotary Position Embeddings (RoPE) as described in (Su et al., 2021) during the fine-tuning phase. Sequence parallelism is implemented for the training at the 262,144 context length. We use low-rank adaptation for context extension fine-tuning (Chen et al., 2024b). These processes collectively require approximately 336 GPU hours on machines equipped with 80GB A100 GPUs.

## 3.3    STAGE5: LONG SUPERVISED FINE-TUNING

**Long video instruction following** To facilitate the fine-tuning of long videos, we constructed a new, dedicated dataset for long video training, each consisting of 15,292 videos. We use the original long videos from the Shot2Story dataset (Han et al., 2023). Each video includes different questions and answers: one for generating captions and another for answering questions, enabling diverse applications in video understanding. Figure 5 illustrates the process for generating instruction-following datasets from long videos. Initially, the long video is segmented into shorter clips, each approximately 10 seconds in duration. These clips are then independently annotated with descriptive captions utilizing the VILA-1.5 model. Subsequently, an LLM is employed to generate question-and-answer pairs derived from the captions of these clips. The generated questions encompass summarization and other queries relevant to the comprehensive understanding of long video content.

As in Figure 4, the left chart categorizes videos into several domains, including Travel & Events, Sports, Education, Pets & Animals, People & Blogs, News & Politics, Music, Science & Technology, Comedy, Entertainment, Film, and Gaming, ensuring a wide-ranging representation of video content. The right chart breaks down the categories of questions into Spatial, Attribute, Action, Object, OCR, Synopsis, and Temporal, reflecting the variety of inquiries and cognitive tasks that

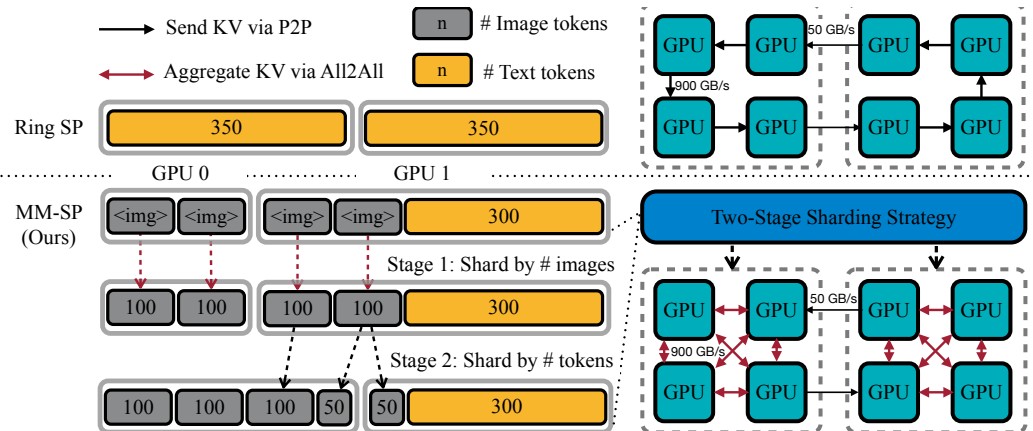

Figure 6: Sharding strategy and communication pattern of MM-SP. For sharding strategy, Ring SP is designed for text-only modalities, without optimization for the workload of an image encoder. Our MM-SP implements a novel sharding strategy that balances the computational load between the image encoder and the language modeling stages. For communication pattern, Ring SP (Liu et al., 2023a; Li et al., 2023a) (top) relies on P2P communication for both intra-node and inter-node settings, resulting in underutilization of intra-node bandwidth. MM-SP (bottom) adopts 2D-Attention (Fang & Zhao, 2024; Gu et al., 2024) mechanism which utilizes intra-node All-to-All (All2All) and inter-node Point-to-Point (P2P) communication to transfer keys and values (KV), enhancing the efficiency of intra-node NVLink utilization. The bandwidth is for H100.

the dataset can address. This dataset provides a rich resource for advancing the understanding and processing of long video formats in supervised fine-tuning.

Once we acquired the long video dataset, applying it for supervised fine-tuning introduced new challenges, primarily due to the substantial number of frames in each sample—often ranging in the hundreds or even thousands. For instance, a single sequence from 1400 video frames can encompass around 274k tokens. Existing data-parallel training systems struggle to handle such extensive contexts. We developed the MM-SP system (Section 4) to efficiently train long-context VLMs.

# 4 MULTI-MODAL SEQUENCE PARALLELISM

Training long-context Vision-Language Models (VLMs) results in substantial memory demands. The most widely used open-source solution, fully sharded data parallelism, does not distribute the activations generated by a single sequence, making it unsuitable for our needs. Consequently, we developed a custom system based on sequence parallelism (Li et al., 2021; 2023a; Liu et al., 2023a; Jacobs et al., 2023), a technique commonly employed in existing foundation model systems to optimize text-only LLM training. However, we discovered that existing systems are neither efficient nor scalable enough to handle our long-context VLM workloads.

## 4.1 LIMITATIONS OF EXISTING SYSTEMS

**Modality heterogeneity.** In text-only LLMs, sequences are processed by a single tokenizer into tokens, allowing for straightforward distribution of tokens across multiple GPUs. However, VLMs incorporate an encoder architecture where non-text data is initially represented by a placeholder token (*e.g.,* ) and subsequently encoded into multiple real tokens during training. For instance, a single video frame typically requires around 256 tokens (Lin et al., 2023b). Due to the differing processing requirements of visual and text modalities, a simplistic implementation that treats placeholder tokens the same as text tokens leads to an imbalance in GPU workloads (Figure 6).

**Networking heterogeneity.** Our multi-modality comprises extremely long videos (Figure 1), which requires employing sequence parallelism in a *multi-node* setting. In a multi-node setting, inter-node and intra-node network bandwidth differs significantly. For example, the NVIDIA DGX H100

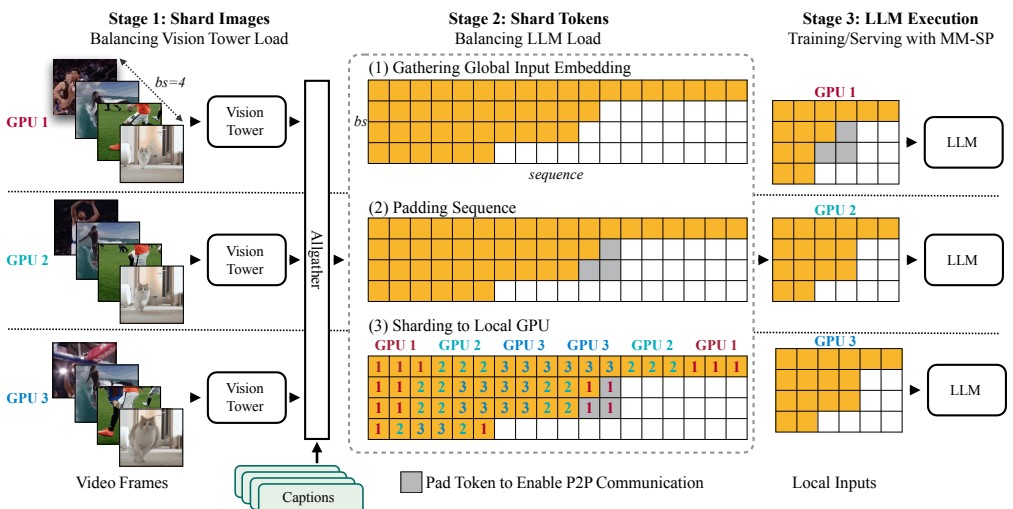

Figure 7: Workflow of Multi-Modal Sequence Parallelism with a Batch Size (bs) of 4 and a Sequence Parallel Size (SP_Size) of 3. To accomodate multi-modal inputs, we developed a customized sharding strategy that ensures balanced workload distribution and compatibility with SP communication.

utilizes NVLink at 900 GB/s for intra-node GPU communication and InfiniBand at 50 GB/s for inter-node GPU communication (single path), resulting in an 18× difference in bandwidth. Previous work, Ring-Style sequence parallelism (Li et al., 2021; 2023a; Liu et al., 2023a; Zhu, 2023) ignores the heterogeneous networking feature on GPUs and utilizes P2P communication in both inter-node and intra-node settings. This design induces excessive communication costs where they usually attempt to overlap them into computation. However, we found that this design cannot always hide the overhead, and even slows down the computation kernel (Table 2).

**Limited maximal sequence length.** DeepSpeed-Ulysses (Jacobs et al., 2023) presents a potential solution to the communication challenges in ring-style sequence parallelism by employing All-to-All communication primitives, which reduce the overall communication volume. However, this approach has its limitations. The design relies on parallelizing along the attention head dimension rather than the sequence dimension during attention computation. As a result, DeepSpeed-Ulysses cannot scale effectively beyond the number of attention heads. For instance, the Llama-3 8B model uses Grouped Query Attention (GQA) with 8 Key-Value heads, which restricts the maximum sequence parallelism degree to 8. Even when using replication for Key-Value heads, which introduces additional communication overhead (Li et al., 2023a), the highest achievable sequence parallelism degree is still limited to 32 (the number of Query heads). This constraint is insufficient for handling extremely long sequences, such as full-length movies.

## 4.2 Multi-Modal Sequence Parallelism Training Mode

After identifying the limitations in existing systems, we conclude that an ideal multi-modal sequence parallelism approach should prioritize efficiency and scalability by addressing both modality and network heterogeneity, and should also be capable of scaling beyond the number of attention heads. To achieve this, we adopt 2D-attention (Fang & Zhao, 2024; Gu et al., 2024) mechanism for sequence parallelism. For instance, as illustrated on the left in Figure 6, to enable an 8-degree sequence parallelism across 2 nodes, we construct a 4×2 communication mesh using 2D-SP. In this setup, the A2A process group, with a size of 4, distributes the QKV tensors according to the head dimension and re-partitions them along the sequence dimension within each node. Simultaneously, the P2P process group, with a size of 2, transfers the partitioned KV chunks between nodes. Additionally, to further explain how the 2D-attention mechanism operates, we depict the attention computation schedule using different methods in Figure 11.

**MM-SP workflow.** To address the challenge of modality heterogeneity, we propose a two-stage sharding strategy that optimizes the compute workload for both image encoding and language mod-

Table 2: The forward and backward attention kernel wall-clock time with or without the overlapping design (Unit: $\mu$s). The communication overlap design in Ring-style SP **slows down** the attention kernel by occupying streaming multiprocessor (SM) resources.

| Seq. length | 4K | 8K | 16 K | 24K | 32K |
|---|---|---|---|---|---|
| forw. w/o | 29.5 | 49.3 | 122.1 | 239.2 | 402.9 |
| forw. w/ | 35.0 (+18.6%) | 54.6 (+10.7%) | 131.2 (+7.5%) | 250.9 (+4.8%) | 420.1 (+4.2%) |
| backw. w/o | 77.7 | 123.3 | 362.9 | 730.0 | 1218.9 |
| backw. w/ | 82.2 (+5.8%) | 129.8 (+5.3%) | 367.0 (+1.1%) | 743.2 (+1.8%) | 1225.3 (+0.5%) |

eling stages. As illustrated in Figure 7, the process begins by evenly distributing images (e.g., video frames) across devices within the sequence parallelism (SP) process group, thereby achieving load balancing during the image encoding stage. In the second stage, we aggregate global vision and text inputs for token-level sharding. To support ring-based attention, sequences are extended with arbitrary dummy tokens, ensuring that each sequence can be evenly divided according to the ring-based SP degree. This adjustment maintains consistency with the original approach by modifying label inputs to ignore padded tokens during loss calculation. We implement a balanced sharding strategy that distributes the context to each rank from both ends, ensuring equal computation across ranks. The effectiveness of this strategy will be demonstrated later (Table 5). Since this redistribution is performed only once during training, the overhead is minimal. Finally, the balanced local inputs are processed by the LLM backbone, utilizing 2D-Attention to achieve efficient sequence parallelism.

### 4.3 MULTI-MODAL SEQUENCE PARALLELISM INFERENCE MODE

The model we developed through sequence parallelism training is capable of handling long-context multi-modal downstream tasks. However, the most commonly used inference system, built on HuggingFace Transformers, typically operates on a single GPU. This lack of distributed implementation limits the maximum sequence length that can be processed during inference. The most straightforward solution within HuggingFace Transformers is to use its pipeline parallelism inference feature, which shards a single model across multiple devices on a layer-by-layer basis (Huang et al., 2019; Narayanan et al., 2019). However, this approach is inefficient, as it only activates one GPU at a time. Additionally, it struggles to support long sequence because the first device must store large input embeddings and images, creating a memory bottleneck.

To address these limitations, we implemented sequence parallelism for distributed inference in VLMs. Unlike the training mode, the inference system additionally manages tensors, such as input tokens and position encodings, that progressively change during the decoding phase (Yu et al., 2022). It detects signals from the machine with the last token to terminate the distributed process appropriately. Compared to HuggingFace's pipeline parallelism strategy, our inference mode is more efficient, as all devices participate in computation simultaneously, accelerating the process by a factor proportional to the number of machines (Figure 8). Furthermore, it is scalable, with memory evenly distributed across devices, enabling longer sequences with additional machines.

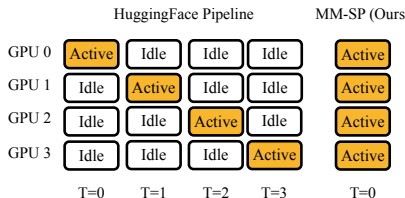

Figure 8: Inference scheduling comparison between HuggingFace Pipeline and MM-SP, illustrated with 4 GPUs. MM-SP utilizes all GPUs concurrently.

## 5 EXPERIMENTAL RESULTS

### 5.1 TRAINING AND INFERENCE SYSTEM

Our training and inference systems can be integrated with HuggingFace Transformers through straightforward monkey patching, in line with the popular open-source approach outlined in (Zheng et al., 2023). In this section, we present a quantitative evaluation of the training system's throughput, the inference system's latency, and the maximum supported sequence length.

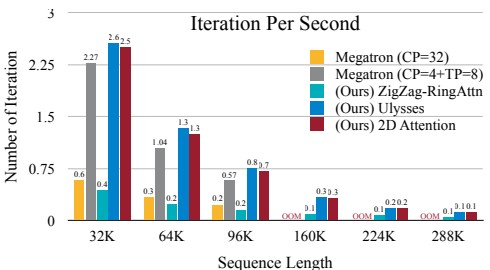 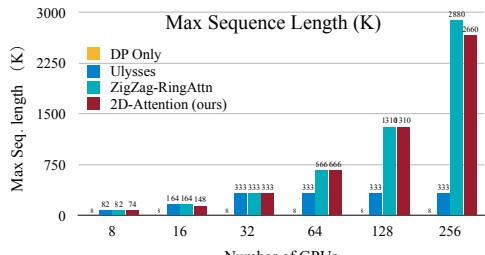

Figure 9: Performance comparison of training systems on 32 H100 GPUs. MM-SP is as scalable as ZigZag-RingAttn, and as efficient as Ulysses.

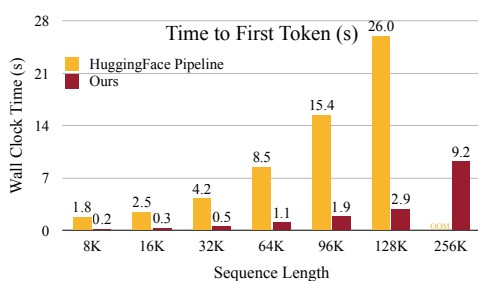 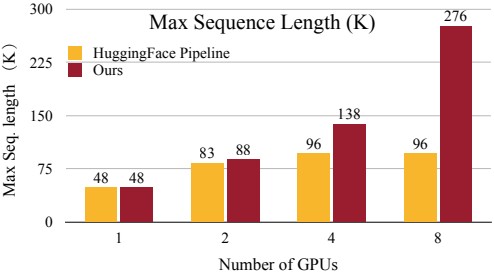

Figure 10: Performance comparison of inference systems on 8 H100 GPUs.

### 5.1.1 TRAINING SYSTEM

**Baselines and hardware setup** For training efficiency, we compare our system with ZigZag ring-style sequence parallelism, which incorporates load balancing and GPU optimization (ZIGZAG-RINGATTN for consistency)(Li et al., 2023a; Zhu, 2023; Liu et al., 2024a; Korthikanti et al., 2023). We use a widely adopted open-source implementation (Zhu, 2023). To reduce the memory footprint of models, gradients, and optimizer states, we employ Fully-Sharded Data Parallelism (FSDP)(Zhao et al., 2023) instead of Zero-3(Rajbhandari et al., 2020) (Table 7). Additionally, we compare our system with the expert-designed and highly optimized Megatron-LM(Shoeybi et al., 2019; Korthikanti et al., 2023) system, focusing on their implementation of sequence parallelism, termed "context parallelism" (CP). We also evaluate a hybrid strategy that combines tensor model parallelism (TP) within a node and CP across nodes, as recommended by the Megatron-LM team for advanced usage.

We conduct most experiments on H100 nodes, each equipped with 8xH100 (80GB) GPUs interconnected via intra-node NVLink and 400 Gbps inter-node InfiniBand. For experiments involving the maximum supported sequence length during training, we extend the setup to 32 A100 nodes, each with 8xA100 (80GB) GPUs, where the conclusions are consistent with those for H100 due to the equivalent total memory. Our evaluations are based on an 8B model with a batch size of 1. Since the Megatron-LM baseline does not natively support VLM training and the visual encoder is typically orders of magnitude smaller than LLMs, we report the main results for the LLM backbone without the visual encoder. An ablation study of the visual encoder is provided in § 5.1.3.

**Throughput** Figure 9 presents throughput results measured as iteration per second over 32 H100 GPUs. These results were obtained after 10 warmup iterations and averaged over 5 iterations to minimize variance. Our system achieves a speedup of $2.1\times$ to $5.7\times$ compared to ZIGZAG-RINGATTN, and performs on par with DeepSpeed-Ulysses. When compared to the more optimized ring-style sequence parallelism in Megatron-LM CP, our method shows a $3.1\times$ to $4.3\times$ speedup. This highlights that our system design effectively addresses the issues inherent in ring-style sequence parallelism, as in § 4.2. Furthermore, our system achieves a $1.1\times$ to $1.4\times$ speedup compared to Megatron-LM's hybrid strategy. Note that our system is currently implemented in Triton (Tillet et al., 2019), and further porting it to C++ could yield even greater speedup. Additionally, we observed that the Megatron-LM system supports a significantly lower maximum sequence length, which is why its results are not in the next section. We observe similar observations using 8 H100 nodes (Table 8).

Table 3: Performance comparisons on 9 video benchmarks, including ActivityNet-QA (Yu et al., 2019), EgoSchema (Mangalam et al., 2023), EventBench (Du et al., 2024), LongVideoBench (Wu et al., 2024), PerceptionTest (Patraucean et al., 2023), MVBench (Li et al., 2024b), NExT-QA (Xiao et al., 2021), VNBench (Zhao et al., 2024), and VideoMME (Fu et al., 2024a).

| Model | LLM Size | ActNet-QA | EgoSchema | EventBench | LVideoBench | PercepTest | MVBench | NExT-QA | VNBench | VideoMME | |
|---|---|---|---|---|---|---|---|---|---|---|---|
| | | test | test | val | val | val | test | mc | val | w/o sub. | w/ sub. |
| GPT-4V | - | 57.0 | - | 32.6 | 61.3 | - | 43.5 | - | - | 59.9 | 63.3 |
| GPT-4o | - | - | - | 53.3 | 66.7 | - | - | - | 64.4 | 71.9 | 77.2 |
| Gemini-1.5-Pro | - | 57.5 | 72.2 | 43.2 | 64.0 | - | - | - | 66.7 | 75.0 | 81.3 |
| Video-LLaVA | 7B | 45.3 | 38.4 | 5.9 | 37.6 | - | 43.5 | - | 12.4 | 39.9 | 41.6 |
| Flash-VStream | 7B | 51.9 | - | - | - | - | - | 61.6 | - | - | - |
| ShareGPT4Video | 8B | 50.8 | - | - | 41.8 | - | 51.2 | - | - | 39.9 | 43.6 |
| VideoLLaMA2 | 7B | 50.2 | 51.7 | 6.9 | - | 51.4 | 54.6 | - | 4.5 | 47.9 | 50.3 |
| VideoLLaMA2.1 | 7B | 53.0 | 53.1 | - | - | 54.9 | 57.3 | - | - | 54.9 | 56.4 |
| Kangaroo | 8B | - | 62.7 | - | 54.8 | - | 61.1 | - | - | 56.0 | 57.6 |
| PLLaVA | 7B | 56.3 | - | 28.2 | 39.2 | - | 46.6 | - | - | - | - |
| LLaVA-OV | 7B | 56.7 | 60.1 | - | 56.4 | 57.1 | 56.7 | 79.4 | 51.8 | 58.2 | 61.5 |
| LongVILA | 7B | **59.5** | **67.7** | **58.0** | **57.1** | **58.1** | **67.1** | **80.7** | **63.0** | **60.1** | **65.1** |

We evaluate the maximum sequence length supported by a fixed number of GPUs by progressively increasing the per-GPU sequence length from 1k to 10k until an out-of-memory error occurs. The results are summarized in Figure 9. To ensure a fair comparison, activation checkpointing is disabled. Vanilla data parallelism fails to scale for long videos at larger cluster sizes. DeepSpeed-Ulysses partitions based on attention heads, which limits its ability to scale to higher context lengths, as the 8B model has only 32 attention heads. Consequently, our approach supports approximately $8\times$ higher context lengths when scaled to 256 GPUs. Additionally, our system achieves a similar context length scaling as ZIGZAG-RINGATTN, with support for over **2 million** context length on 256 GPUs.

In summary, our training system combines the best of both worlds—it achieves scalability comparable to ZIGZAG-RINGATTN while maintaining the throughput of DeepSpeed-Ulysses. Additionally, it delivers a $1.3\times$ speedup and supports $2.5\times$ longer context lengths compared to the highly optimized Megatron-LM.

### 5.1.2 INFERENCE SYSTEM

We evaluated our inference system against HuggingFace Pipeline parallelism using a single node with 8 H100 GPUs and the 8B model (Figure 10). Our system achieves an $8.2\times$ speedup compared to HuggingFace Pipeline on 8xH100 GPUs. This significant improvement is primarily due to HuggingFace Pipeline inference activating only one GPU at a time, whereas our method leverages all GPUs to compute jointly. Figure 10 compares the maximum supported sequence length, where our method supports sequences that are $2.9\times$ longer than those supported by HuggingFace Pipeline. Specifically, during 96K sequence length inference, HuggingFace Pipeline stores 80GB of activations on the first GPU and only 18GB on the remaining GPUs. This imbalanced allocation of activations limits the maximum supported sequence length.

### 5.1.3 EFFECT OF TWO-STAGE SHARDING

We evaluate the impact of our two-stage sharding strategy using a video captioning dataset (Chen et al., 2024a). We compare our two-stage sharding to a one-stage baseline that only distributes workload based on the number of images. We measure the time per iteration across different numbers of H100 GPUs. For $k$ GPUs, we use $k$ images per video and a batch size of $k$. The results, shown in Table 5, indicate a speedup ranging from 1% to 7%. This improvement is primarily observed

Table 4: Performance comparison on VideoMME (Fu et al., 2024a) benchmark in details.

| Model | LLM Size | Frames | w/o subtitle | | | | w subtitle | | | |
|---|---|---|---|---|---|---|---|---|---|---|
| | | | Overall | Short | Medium | Long | Overall | Short | Medium | Long |
| Video-LLaVA | 7B | 8 | 39.9 | 45.3 | 38.0 | 36.2 | 41.6 | 46.1 | 40.7 | 38.1 |
| SliME | 8B | 8 | 45.3 | 53.3 | 55.4 | 39.8 | 47.2 | 55.4 | 44.4 | 41.7 |
| ShareGPT4Video | 8B | 16 | 39.9 | 48.3 | 36.3 | 35.0 | 43.6 | 53.6 | 39.3 | 37.9 |
| VideoChat2 | 7B | 16 | 39.5 | 48.3 | 37.0 | 33.2 | 43.8 | 52.8 | 39.4 | 39.2 |
| VideoLLaMA2 | 7B | 16 | 47.9 | 56.0 | 45.4 | 42.1 | 50.3 | 59.4 | 47.6 | 43.8 |
| Chat-Univi-v1.5 | 7B | 64 | 40.6 | 45.7 | 40.3 | 35.8 | 45.9 | 51.2 | 44.6 | 41.8 |
| Kangaroo | 8B | 64 | 56.0 | 66.1 | 55.3 | 46.7 | 57.6 | 68.0 | 55.4 | 49.3 |
| ShareGemini | 7B | 64 | 43.2 | 49.1 | 41.3 | 39.1 | 47.9 | 49.1 | 47.3 | 43.4 |
| LongVA | 7B | 128 | 52.6 | 61.1 | 50.4 | 46.2 | 54.3 | 61.1 | 53.6 | 47.6 |
| InternVL-V1.5 | 20B | 10 | 50.7 | 60.2 | 46.4 | 45.6 | 52.4 | 61.7 | 49.1 | 46.6 |
| VITA | 8x7B | 20 | 55.0 | 64.2 | 53.3 | 47.6 | 57.6 | 67.9 | 55.3 | 49.6 |
| Video-CCAM | 14B | 96 | 53.9 | 62.1 | 52.8 | 47.0 | 56.1 | 63.9 | 55.9 | 48.3 |
| LongVILA | 1.5B | 256 | 53.6 | 66.2 | 49.3 | 45.3 | 57.5 | 70.2 | 54.1 | 48.2 |
| | 7B | | **60.1** | **69.0** | **58.3** | **53.0** | **65.1** | **72.9** | **64.9** | **57.4** |

in longer captioning tasks, where the baseline suffers from workload imbalance due to the lack of sharding based on the number of text tokens.

## 5.2 GENERAL VIDEO UNDERSTANDING

Table 3 presents the performance of LongVILA, comparing to state-of-the-art models (Lin et al., 2023a; Zhang et al., 2024d; Chen et al., 2024a; Li et al., 2023c; Cheng et al., 2024; Jin et al., 2023; Zhang et al.; Share, 2024; Chen et al., 2024c; Fu et al., 2024b; Fei et al., 2024; Liu et al., 2024c; Xu et al., 2024; Li et al., 2024a; Zhang et al., 2024a) on popular video benchmarks. LongVILA-7B achieves strong performance across all these benchmarks. Table 4 compares their effectiveness across short, medium, and long video lengths, as well as overall performance on VideoMME. LongVILA, utilizing 256 frames, achieves an overall score of 60.1 / 65.1 without / with subtitle, which are competitive results. We include LongVILA-1.5B (starting from Qwen2-1.5B (qwe, 2024)) for evaluation, which is also competitive. We provide a detailed model complexity of LongVILA among various model size, number of frames, context length, latency and FLOPs in Table 10 in the appendix. We also do the ablation on training schedules in Table 1.

## 5.3 NEEDLE-IN-A-HAYSTACK

In Figure 2, we present the results of the Needle in a Haystack experiment for long videos. Following the methodology established in the existing literature (Zhang et al.), we prepared a long video and sampled a fixed number of frames. We inserted specifically designed images at various depths and tasked the model with answering corresponding questions. The 32-frame baseline model (left) was unable to accurately retrieve the correct images beyond 32 frames. In contrast, the LongVILA model (right) demonstrated 99.8% accuracy across 6,000 frames, which contains more than 1 million tokens. To our best knowledge, this is the first VLM which can handle 1 million context length.

## 6 CONCLUSION

We introduce LongVILA, a comprehensive full-stack solution for long-context visual language models, encompassing model training pipeline and distributed system. Based on our curated long video datasets and five-stage training pipeline, our LongVILA model extends the feasible frame count from 8 to 2048, precisely capturing fine-grained information from 2-hour needle-in-a-haystack videos, with more than 1 million tokens Our LongVILA-7B model achieves strong performance across popular video benchmarks, especially on VideoMME, *e.g.,* 65.1% accuracy with subtitle. Our system efficiently scales context length up to 2 million tokens, achieving speedups of 2.1× to 5.7× compared to ring sequence parallelism and 1.1× to 1.4× compared to a hybrid Megatron context and tensor parallelism.

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

# A APPENDIX

## A.1 LONGVILA-CAPTION

We have developed a long video captioning benchmark, LongVILA-Caption, consisting of 100 long videos, with captions generated as detailed in Section 3.3, and verified through human examination. In line with the methodology of VideoChatGPT (Maaz et al., 2024), we evaluate the predictions of each model based on their correctness, detailed orientation, and contextual understanding. For instance, we assess correctness by employing GPT-4 to predict scores using a specific prompt. Additionally, we present two examples in Figures 13 and 14, featuring long videos in sports and technology. These examples demonstrate that LongVILA, with its capability to process more frames, offers a more comprehensive understanding of videos compared to its short-frame counterpart.

The Table 6 presents the performance metrics for the LongVILA models being trained and evaluated on varying numbers of frames: 8, 128, and 256. As the number of frames increases, the model's performance improves significantly. Specifically, the average scores rise from 2.00 to 3.26, highlighting the model's enhanced capability in generating accurate and rich captions with more frames.

Table 5: Iteration time (seconds) on the dataset (Chen et al., 2024a) with and without our two-stage sharding strategy.

|  | 2 GPUs | 4 GPUs | 8 GPUs |
|---|---|---|---|
| one-stage | 0.78 | 0.89 | 1.20 |
| two-stage | 0.77 | 0.86 | 1.12 |

Table 6: Evaluation of LongVILA-Caption performance across different frame counts.

| Frames | Correctness | Detailed | Contextual | Average |
|---|---|---|---|---|
| 8 | 1.87 | 1.85 | 2.27 | 2.00 |
| 128 | 2.36 | 2.44 | 2.79 | 2.53 |
| 256 | 3.23 | 3.11 | 3.43 | 3.26 |

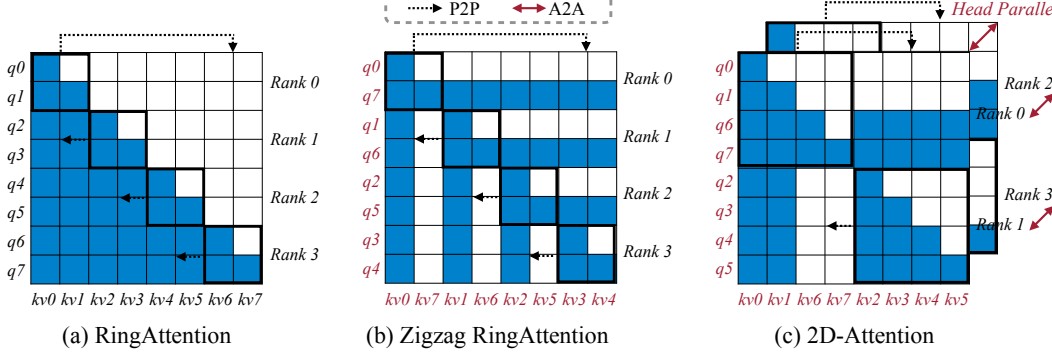

(a) RingAttention     (b) Zigzag RingAttention     (c) 2D-Attention

Figure 11: Comparison of RingAttention Liu et al. (2023a), ZIGZAG-RINGATTN Zhu (2023), and 2D-Attention (Fang & Zhao, 2024). The blue block indicates communication between QKV, while the black frame represents local attention computation within each SP group rank. The sequence length is 8 and the global SP degree is 4. Due to the triangular structure of causal attention computations, RingAttention experiences a computation imbalance, where rank 0 becomes idle after the first round while rank 3 continues computing through all stages. ZIGZAG-RINGATTN addresses this by reordering input sequence tokens along the sequence dimension to achieve load balance. The 2D-Attention mechanism uses a ring parallel degree of 2 and a head parallel degree of 2, resulting in an effective global sequence parallel degree of 4. This approach also incorporates a workload balancing strategy within the ring-based process group and uses the All-to-All operation to distribute QKV tensors across devices based on the head dimension, ensuring efficient and balanced computation.

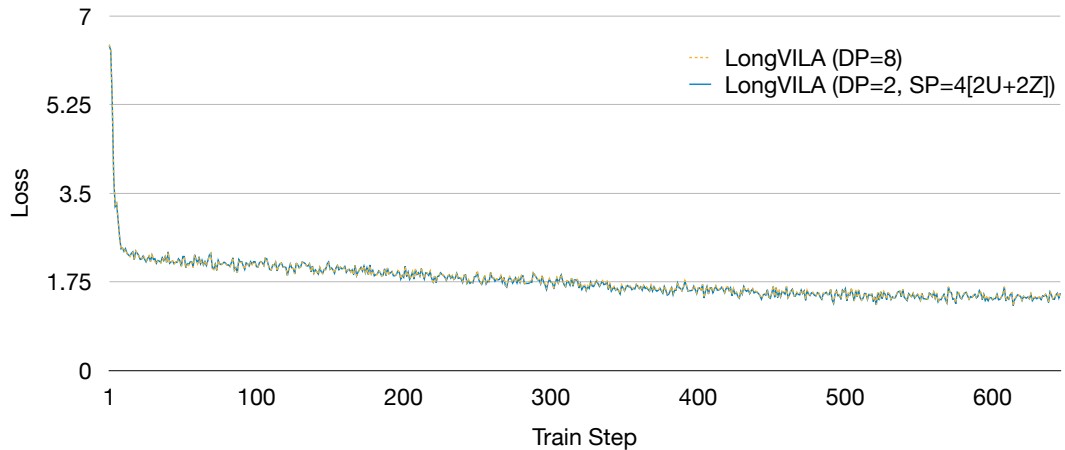

Figure 12: Convergence evaluation. We compare the training loss curves for LongVILA with and without sequence parallelism. This figure illustrates the convergence of the LongVILA model on 8 H100 GPUs, with a sequence parallelism degree of 4, compared to pure data parallelism training. 2U+2Z indicates that 2D-Attention mechanism is enabled, where both the Ulysses and Zigzag-Ringttn degrees are 2. The training dataset is shot2story. The two curves align closely, indicating that our MM-SP system does not negatively impact training quality.

Table 7: Our training system, used in conjunction with FSDP (Zhao et al., 2023) or Zero-3 (Rajbhandari et al., 2020), on 32 H100 GPUs. We found that FSDP offers more efficient memory management, which led us to select it as our default configuration. (Time per iteration, seconds).

| Sequence Length | Zero-3 | | | FSDP | | |
|---|---|---|---|---|---|---|
| | ZIGZAG-RINGATTN | Ulysses | 2D Attention | ZIGZAG-RINGATTN | Ulysses | 2D Attention |
| 320 K | OOM | OOM | OOM | 23.57 | 10.70 | 11.12 |
| 288 K | OOM | OOM | OOM | 20.24 | 8.68 | 8.65 |
| 256 K | OOM | OOM | OOM | 17.54 | 6.98 | 7.04 |
| 224 K | 19.04 | 7.06 | 5.73 | 15.22 | 5.47 | 5.53 |
| 192 K | 13.01 | 4.24 | 4.38 | 12.97 | 4.15 | 4.24 |
| 160 K | 10.73 | 3.09 | 3.23 | 10.83 | 3.02 | 3.11 |
| 128 K | 8.63 | 2.16 | 2.30 | 8.38 | 2.07 | 2.17 |
| 96 K | 6.49 | 1.43 | 1.53 | 6.35 | 1.33 | 1.41 |
| 64 K | 4.40 | 1.01 | 1.08 | 4.25 | 0.76 | 0.80 |
| 32 K | 2.06 | 1.58 | 1.04 | 2.26 | 0.39 | 0.40 |

Table 8: Training system throughput comparison on 64 H100 GPUs, measured in time per iteration (seconds). Ulysses is not included in this comparison as it supports only up to 32 GPUs.

| Sequence length | Megatron-LM | | Ours | | |
|---|---|---|---|---|---|
| | CP | CP=8+TP=8 | ZIGZAG-RINGATTN | Ulysses | 2D Attention |
| 640 K | OOM | OOM | 88.4 | - | OOM |
| 578 K | OOM | OOM | 77.2 | - | 16.9 |
| 512 K | OOM | OOM | 66.1 | - | 13.31 |
| 448 K | OOM | OOM | 57.5 | - | 10.39 |
| 384 K | OOM | OOM | 48.6 | - | 7.80 |
| 320 K | OOM | OOM | 40.5 | - | 5.63 |
| 256 K | OOM | 5.31 | 32.2 | - | 3.93 |
| 192 K | 8.81 | 3.10 | 24.1 | - | 2.49 |
| 128 K | 7.10 | 1.57 | 16.0 | - | 1.36 |
| 64 K | 3.09 | 0.61 | 8.04 | - | 0.57 |
| 32 K | 1.86 | 0.44 | 4.24 | - | 0.33 |

Table 9: Comparison with state-of-the-art methods (Li et al., 2023b; Dai et al., 2023; Bai et al., 2023; Liu et al., 2023b; Lin et al., 2023b; Liu et al., 2024b; Tong et al., 2024; Li et al., 2024c) on 10 image based VLM benchmarks. S3 refers to the stage 3 model in LongVILA training pipeline.

| Method | LLM | Res. | VQA$^{v2}$ | GQA | VizWiz | SQA$^{I}$ | VQA$^{T}$ | MMB | MMB$^{CN}$ | SEED | LLaVA$^{W}$ | MM-Vet |
|---|---|---|---|---|---|---|---|---|---|---|---|---|
| BLIP-2 | Vicuna-13B | 224 | 41.0 | 41 | 19.6 | 61 | 42.5 | – | – | 46.4 | 38.1 | 22.4 |
| InstructBLIP | Vicuna-7B | 224 | – | 49.2 | 34.5 | 60.5 | 50.1 | 36 | 23.7 | 53.4 | 60.9 | 26.2 |
| | Vicuna-13B | 224 | – | 49.5 | 33.4 | 63.1 | 50.7 | – | – | – | 58.2 | 25.6 |
| Qwen-VL | Qwen-7B | 448 | 78.8 | 59.3 | 35.2 | 67.1 | 63.8 | 38.2 | 7.4 | 56.3 | – | – |
| Qwen-VL-Chat | Qwen-7B | 448 | 78.2 | 57.5 | 38.9 | 68.2 | 61.5 | 60.6 | 56.7 | 58.2 | – | – |
| LLaVA-1.5 | Vicuna-1.5-7B | 336 | 78.5 | 62.0 | 50.0 | 66.8 | 58.2 | 64.3 | 58.3 | 58.6 | 63.4 | 30.5 |
| | Vicuna-1.5-13B | 336 | 80.0 | 63.3 | 53.6 | 71.6 | 61.3 | 67.7 | 63.6 | 61.6 | 70.7 | 35.4 |
| VILA | Llama 2-7B | 336 | 79.9 | 62.3 | 57.8 | 68.2 | 64.4 | 68.9 | 61.7 | 61.1 | 69.7 | 34.9 |
| | Llama 2-13B | 336 | 80.8 | 63.3 | 60.6 | 73.7 | 66.6 | 70.3 | 64.3 | 62.8 | 73.0 | 38.8 |
| LLaVA-NeXT-8B | Llama 3-8B | 672 | – | 65.2 | – | 72.8 | 64.6 | 72.1 | – | – | 80.1 | – |
| Cambrian-1-8B | Llama 3-8B | 1024 | – | 64.6 | – | 80.4 | 71.7 | 75.9 | – | – | – | – |
| Mini-Gemini-HD-8B | Llama 3-8B | 1536 | – | 64.5 | – | 75.1 | 70.2 | 72.7 | – | – | – | – |
| LongVILA-7B (S3) | Qwen2-7B | dynamic | **85.4** | **65.4** | **65.0** | **98.5** | **77.8** | **83.4** | **80.0** | **70.6** | **77.6** | **51.7** |

Table 10: Detailed model complexity analysis of LongVILA among various model size, number of frames, parameters, latency (ms), and TFLOPs. We profile LongVILA models into 4 types of components. Image Encoder includes vision tower and mm projector. LLM Linears include k/q/v/o projectors and linears in decoder layers. LLM Attention is the attention computation. LLM Others include other components, like embeddings, output heads, and normalization layers. We use fp16 data type, Flash-Attention2 (Dao, 2024) on one A100 GPU for latency measurement. As the number of frames increases, the FLOPs and latency of LLM Attention grow quadratically, whereas other components increase linearly. LLM Attention represents the predominant computational cost in long video understanding, highlighting the MM-SP system.

| Frames | Context | Metric | LongVILA-1.5B | | | | LongVILA-7B | | | |
|---|---|---|---|---|---|---|---|---|---|---|
| | | | Image Encoder | LLM Linears | LLM Attention | LLM Others | Image Encoder | LLM Linears | LLM Attention | LLM Others |
| | | Params | 0.44B | 1.31B | - | 0.23B | 0.46B | 6.53B | - | 1.09B |
| 32 | 6415 | Latency | 196.0 | 109.3 | 27.0 | 28.9 | 201.0 | 426.2 | 51.1 | 42.1 |
| | | TFLOPs | 5.13 | 4.20 | 1.77 | 0.75 | 5.22 | 20.93 | 4.13 | 1.74 |
| 64 | 12719 | Latency | 375.2 | 208.1 | 80.7 | 50.9 | 375.8 | 831.1 | 173.8 | 80.8 |
| | | TFLOPs | 10.26 | 8.33 | 6.96 | 1.48 | 10.45 | 41.50 | 16.24 | 3.46 |
| 128 | 25327 | Latency | 740.4 | 403.2 | 288.1 | 97.1 | 755.1 | 1642.4 | 644.9 | 158.3 |
| | | TFLOPs | 20.52 | 16.60 | 27.59 | 2.95 | 20.89 | 82.64 | 64.38 | 6.89 |
| 256 | 50543 | Latency | 1456.0 | 811.2 | 1087.7 | 192.5 | 1476.3 | 3308.2 | 2529.6 | 325.1 |
| | | TFLOPs | 41.05 | 33.12 | 109.89 | 5.88 | 41.78 | 164.92 | 256.40 | 13.74 |
| 512 | 100975 | Latency | 2921.0 | 1660.9 | 4359.2 | 389.4 | 2980.2 | 6675.1 | 10149.3 | 653.4 |
| | | TFLOPs | 82.09 | 66.17 | 438.59 | 11.75 | 83.57 | 329.48 | 1023.37 | 27.45 |

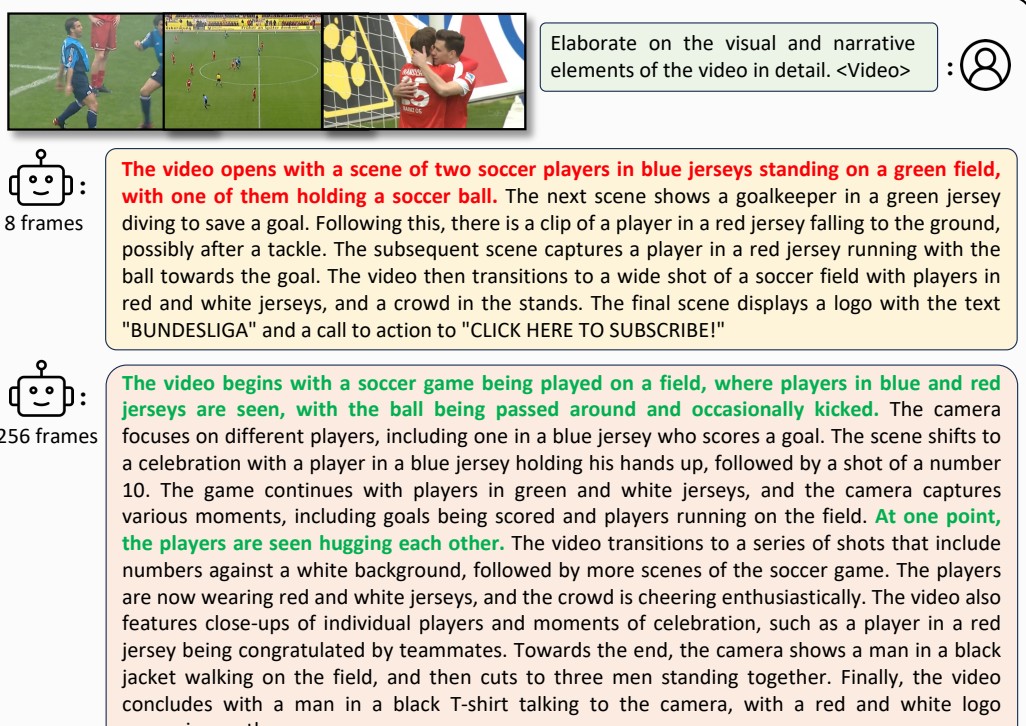

Figure 13: Examples of sports long video caption with LongVILA. For the gameplay opening, the 8-frame baseline describes only static image, two players in only blue jerseys. In contrast, 256-frame LongVILA describes players in blue and red jerseys passing and kicking the ball. In addition, the 256-frame version also include the detail of players hugging emphasizes the celebratory aspects, which is missing in the 8-frame baseline.

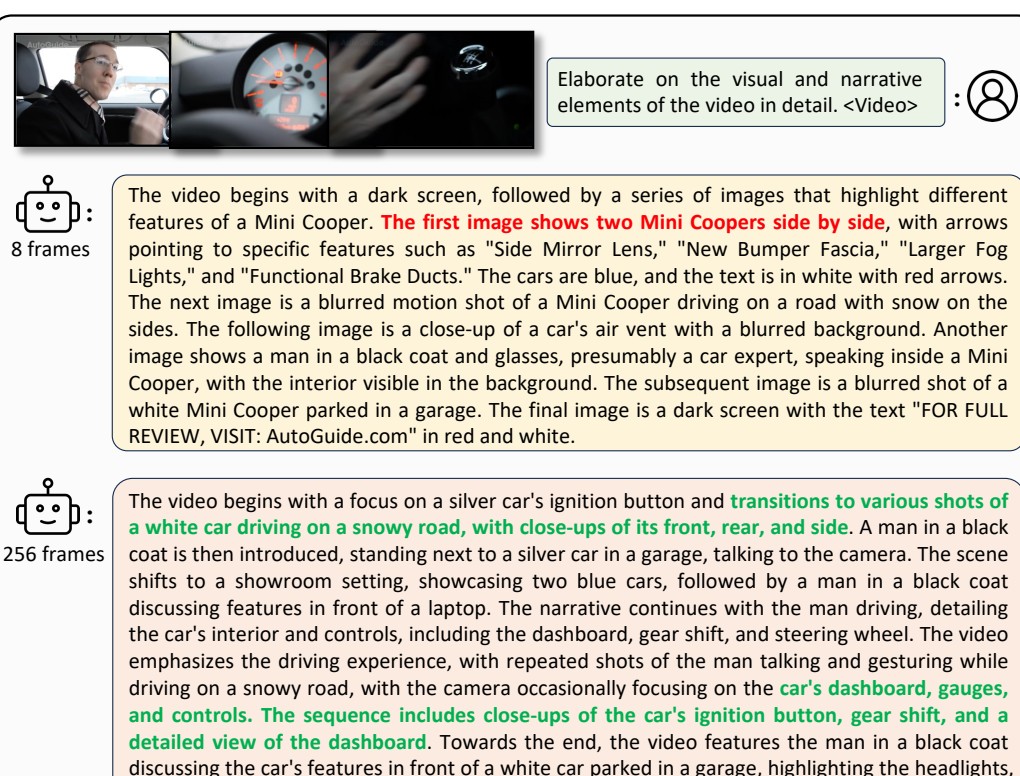

Figure 14: Examples of technology long video caption with LongVILA. At the beginning of captions, the 8-frame baseline only describes static image and two cars. In contrast, the 256-frame LongVILA describes the car on snowy road, covering front, rear, and side views. For details, the 256-frame LongVILA describes close-ups of ignition button, gear shift, and dashboard elements, which are missing in the 8-frame baseline.

