# OpenReview forum: "LongVILA: Scaling Long-Context Visual Language Models for Long Videos"
_ICLR.cc/2025/Conference — ICLR 2025 Poster_

### Official Review · Reviewer_MTsu · 2024-11-03

**Soundness:** 3
**Presentation:** 3
**Contribution:** 3
**Rating:** 6
**Confidence:** 5

**Summary:**

This paper proposes LongVILA, a solution for long-context visual-language models. The authors incorporate two additional stages, i.e., long context extension and long video supervised fine-tuning. A long-context multi-modal sequence parallelism system is proposed to efficiently parallelize long video training and inference.

**Strengths:**

1. This paper implements a five-stage training curriculum: multi-modal alignment, large-scale pre-training, short supervised fine-tuning, context extension for LLMs, and long supervised fine-tuning.
2. An efficient and user-friendly framework (multi-modal sequence parallelism) is proposed to support training and inferencing
memory-intensive long-context VLMs.
3. The presentations are good.

**Weaknesses:**

1. I suggest the authors to involve more baselines in Table 3. For example, PLLaVA, Flash-VStream, Video-LLaMA-2.
2. This paper has not reported results on some (long) video QA benchmarks, for example, MoVQA, ActivityNet-QA, Ego-Schema, MV-Bench. I suggest the authors to include them.
3. Is there any model complexity analysis? For example, #Params, GFLOPs.


[1] PLLaVA : Parameter-free LLaVA Extension from Images to Videos for Video Dense Captioning

 [2] VideoLLaMA 2: Advancing Spatial-Temporal Modeling and Audio Understanding in Video-LLMs

[3] Flash-VStream: Memory-Based Real-Time Understanding for Long Video Streams

**Questions:**

See weakness.

---

> ### Author Response · Authors · 2024-11-20
> **Response to Reviewer MTsu (Part 1)**
>
> We sincerely appreciate your insightful comments. Below, we provide responses to each concern.
>
> **Q1: “More baselines and video QA benchmarks.”**
>
> **A:** Thank you for the reminder.
>
> Regarding baselines, we included additional strong models for comparison, such as PLLaVA [1], Flash-VStream [2], Video-LLaMA-2 [3], LLaVA One-Vision [4], and proprietary models like GPT-4V [5], GPT-4o [6], and Gemini-1.5-Pro [7].
>
> For benchmarks, we added 8 more benchmarks for evaluation: ActivityNet-QA [8], EgoSchema [9], EventBench [10], LongVideoBench [11], PerceptionTest [12], MVBench [13], NExT-QA [14], and VNBench [15]. Due to the inaccessibility of the [MoVQA repository](https://github.com/OpenGVLab/MoVQA) at this period, we will consider it for future evaluation when their repository is available.
>
> Our LongVILA-7B model demonstrates strong performance across these benchmarks, as presented in **Table 3 in the revision**.
>
> | Model | LLM Size | ActNetQA | EgoSchema | MVBench | PercepTest | NExTQA | LVideoBench | EventBench | VNBench | VideoMME | VideoMME |
> | --- | --- | --- | --- | --- | --- | --- | --- | --- | --- | --- | --- |
> |  |  | test | test | test | val | mc | val | val | val | w/o sub. | w/ sub. |
> | GPT-4V | - | 57.0 | - | 43.5 | - | - | 61.3 | 32.6 | - | 59.9 | 63.3 |
> | GPT-4o | - | - | - | - | - | - | 66.7 | 53.3 | 64.4 | 71.9 | 77.2 |
> | Gemini-1.5-Pro | - | 57.5 | 72.2 | - | - | - | 64.0 | 43.2 | 66.7 | 75.0 | 81.3 |
> |  |  |  |  |  |  |  |  |  |  |  |  |
> | Video-LLaVA | 7B | 45.3 | 38.4 | 43.5 | - | - | 37.6 | 5.9 | 12.4 | 39.9 | 41.6 |
> | Flash-VStream | 7B | 51.9 | - | - | - | 61.6 | - | - | - | - | - |
> | ShareGPT4Video | 8B | 50.8 | - | 51.2 | - | - | 41.8 | - | - | 39.9 | 43.6 |
> | VideoLLaMA2 | 7B | 50.2 | 51.7 | 54.6 | 51.4 | - | - | 6.9 | 4.5 | 47.9 | 50.3 |
> | VideoLLaMA2.1 | 7B | 53.0 | 53.1 | 57.3 | 54.9 | - | - | - | - | 54.9 | 56.4 |
> | Kangaroo | 8B | - | 62.7 | 61.1 | - | - | 54.8 | - | - | 56.0 | 57.6 |
> | PLLaVA | 7B | 56.3 | - | 46.6 | - | - | 39.2 | 28.2 | - | - | - |
> | LLaVA-OV | 7B | 56.7 | 60.1 | 56.7 | 57.1 | 79.4 | 56.4 | - | 51.8 | 58.2 | 61.5 |
> | **LongVILA** | **7B** | **59.5** | **67.7** | **67.1** | **58.1** | **80.7** | **57.1** | **58.0** | **63.0** | **60.1** | **65.1** |

---

> ### Author Response · Authors · 2024-11-20
> **Response to Reviewer MTsu (Part 2)**
>
> **Q2: “Model complexity analysis.”**
>
> **A:** Thank you for the helpful suggestion. We conducted a detailed **model complexity** analysis of LongVILA-1.5B and LongVILA-7B, presented in **Table 10 in the revision**. This analysis includes parameters, FLOPs, and latency (ms) across different numbers of frames, context lengths, and model components. We categorize the LongVILA models into four types of components:
>
> - **Image Encoder:** the vision tower and multi-modal projector.
> - **LLM Linears:** k/q/v/o projectors and linear layers in the decoder.
> - **LLM Attention:** attention computation.
> - **LLM Others:** other components such as embeddings, output heads, and normalization layers.
>
> Latency measurements were performed with **fp16** data type and **Flash-Attention2** [16] on a single A100 GPU. This profiling provides a comprehensive view of the computational demands of LongVILA-1.5B and 7B in various settings.
>
> As the number of frames increases, the FLOPs and latency of LLM Attention grow quadratically, whereas other components increase linearly. LLM Attention represents the predominant computational cost in long video understanding, highlighting the necessity of the Multi-Modal Sequence Parallelism (MM-SP) system.
>
> **The model complexity profile of LongVILA-1.5B**
>
> | Frames | Context | Metric | ImageEncoder | LLM Linears | LLM Attention | LLM Others |
> | --- | --- | --- | --- | --- | --- | --- |
> |  |  | Params | 0.44B | 1.31B | - | 0.23B |
> | 32 | 6415 | Latency | 196.0 | 109.3 | 27.0 | 28.9 |
> |  |  | TFLOPs | 5.13 | 4.20 | 1.77 | 0.75 |
> | 64 | 12719 | Latency | 375.2 | 208.1 | 80.7 | 50.9 |
> |  |  | TFLOPs | 10.26 | 8.33 | 6.96 | 1.48 |
> | 128 | 25327 | Latency | 740.4 | 403.2 | 288.1 | 97.1 |
> |  |  | TFLOPs | 20.52 | 16.60 | 27.59 | 2.95 |
> | 256 | 50543 | Latency | 1456.0 | 811.2 | 1087.7 | 192.5 |
> |  |  | TFLOPs | 41.05 | 33.12 | 109.89 | 5.88 |
> | 512 | 100975 | Latency | 2921.0 | 1660.9 | 4359.2 | 389.4 |
> |  |  | TFLOPs | 82.09 | 66.17 | 438.59 | 11.75 |
>
> **The model complexity profile of LongVILA-7B**
>
> | Frames | Context | Metric | Image Encoder | LLM Linears | LLM Attention | LLM Others |
> | --- | --- | --- | --- | --- | --- | --- |
> |  |  | Params | 0.45B | 6.53B | - | 1.09B |
> | 32 | 6415 | Latency | 201.0 | 426.2 | 51.1 | 42.1 |
> |  |  | TFLOPs | 5.22 | 20.93 | 4.13 | 1.74 |
> | 64 | 12719 | Latency | 375.8 | 831.1 | 173.8 | 80.8 |
> |  |  | TFLOPs | 10.45 | 41.50 | 16.24 | 3.46 |
> | 128 | 25327 | Latency | 755.1 | 1642.4 | 644.9 | 158.3 |
> |  |  | TFLOPs | 20.89 | 82.64 | 64.38 | 6.89 |
> | 256 | 50543 | Latency | 1476.3 | 3308.2 | 2529.6 | 325.1 |
> |  |  | TFLOPs | 41.78 | 164.92 | 256.40 | 13.74 |
> | 512 | 100975 | Latency | 2980.2 | 6675.1 | 10149.3 | 653.4 |
> |  |  | TFLOPs | 83.57 | 329.48 | 1023.37 | 27.45 |
>
> [1] Lin Xu, Yilin Zhao, Daquan Zhou, Zhijie Lin, See-Kiong Ng, Jiashi Feng: PLLaVA : Parameter-free LLaVA Extension from Images to Videos for Video Dense Captioning. 2024
>
> [2] Haoji Zhang, Yiqin Wang, Yansong Tang, Yong Liu, Jiashi Feng, Jifeng Dai, Xiaojie Jin: Flash-VStream: Memory-Based Real-Time Understanding for Long Video Streams. 2024
>
> [3] Zesen Cheng, et. al.: VideoLLaMA 2: Advancing Spatial-Temporal Modeling and Audio Understanding in Video-LLMs. 2024
>
> [4] Bo Li, Yuanhan Zhang, Dong Guo, Renrui Zhang, Feng Li, Hao Zhang, Kaichen Zhang, Yanwei Li, Ziwei Liu, Chunyuan Li: LLaVA-OneVision: Easy Visual Task Transfer. 2024
>
> [5] OpenAI. Gpt-4v. 2023
>
> [6] OpenAI. Hello gpt-4o. 2024
>
> [7] Gemini Team. Gemini 1.5: Unlocking multimodal understanding across millions of tokens of context, 2024.
>
> [8] Zhou Yu, Dejing Xu, Jun Yu, Ting Yu, Zhou Zhao, Yueting Zhuang, Dacheng Tao: ActivityNet-QA: A Dataset for Understanding Complex Web Videos via Question Answering. AAAI 2019
>
> [9] Karttikeya Mangalam, Raiymbek Akshulakov, Jitendra Malik: EgoSchema: A Diagnostic Benchmark for Very Long-form Video Language Understanding. NeurIPS 2023
>
> [10] Yifan Du, Kun Zhou, Yuqi Huo, Yifan Li, Wayne Xin Zhao, Haoyu Lu, Zijia Zhao, Bingning Wang, Weipeng Chen, Ji-Rong Wen: Towards Event-oriented Long Video Understanding. 2024
>
> [11] Haoning Wu, Dongxu Li, Bei Chen, Junnan Li: LongVideoBench: A Benchmark for Long-context Interleaved Video-Language Understanding. 2024
>
> [12] Viorica Patraucean, et. al: Perception Test: A Diagnostic Benchmark for Multimodal Video Models. NeurIPS 2023
>
> [13] Kunchang Li, Yali Wang, Yinan He, Yizhuo Li, Yi Wang, Yi Liu, Zun Wang, Jilan Xu, Guo Chen, Ping Lou, Limin Wang, Yu Qiao: MVBench: A Comprehensive Multi-modal Video Understanding Benchmark. CVPR 2024
>
> [14] Junbin Xiao, Xindi Shang, Angela Yao, Tat-Seng Chua: NExT-QA: Next Phase of Question-Answering to Explaining Temporal Actions. CVPR 2021
>
> [15] Zijia Zhao, Haoyu Lu, Yuqi Huo, Yifan Du, Tongtian Yue, Longteng Guo, Bingning Wang, Weipeng Chen, Jing Liu: Needle In A Video Haystack: A Scalable Synthetic Framework for Benchmarking Video MLLMs. 2024
>
> [16] Tri Dao: FlashAttention-2: Faster Attention with Better Parallelism and Work Partitioning. ICLR 2024

---

> > ### Author Response · Authors · 2024-11-25
> > **Follow-Up on Review and Feedback**
> >
> > Dear Reviewer **MTsu**,
> >
> > We hope this message finds you well.
> >
> > We have carefully addressed all your questions and concerns, including conducting additional experiments as requested, and have provided detailed responses in the rebuttal.
> >
> > As the rebuttal deadline is approaching, we would deeply appreciate it if you could share your updated thoughts or revised assessments based on the rebuttal and paper revision, or do not hesitate to let us know if you have additional questions, and we will respond promptly.
> >
> > Thank you again for your thoughtful review and your invaluable contributions to the quality of this paper.
> >
> > Kind regards,
> >
> > Paper 2006 Authors

---

> > > ### Comment · Reviewer_MTsu · 2024-11-25
> > > **Response to author**
> > >
> > > Thanks for answering my questions. It's a good paper and I raise my score.

---

> > > > ### Author Response · Authors · 2024-11-25
> > > > **Reply to Reviewer MTsu**
> > > >
> > > > Thank you again for your constructive feedback and insightful comments, which have significantly enhanced the quality of the work.

---

### Official Review · Reviewer_7Lyg · 2024-11-03

**Soundness:** 3
**Presentation:** 3
**Contribution:** 3
**Rating:** 8
**Confidence:** 4

**Summary:**

The paper introduces LongVILA, a comprehensive framework aimed at enhancing long-context video understanding capabilities in multi-modal foundation models. To optimize model training for long videos, LongVILA introduces two pivotal stages: long context extension and long video supervised fine-tuning. Considering the computational and memory challenges associated with training on long videos, the authors present the long-context Multi-Modal Sequence Parallelism (MM-SP) system. This system significantly improves training efficiency, enabling the processing of context lengths up to 2 million on 256 GPUs without relying on gradient checkpointing. The results demonstrate a remarkable increase in performance metrics, with the long video captioning score rising from 2.00 to 3.26 and achieving 99.5% accuracy in video needle-in-a-haystack tasks. Additionally, LongVILA shows strong performance on the VideoMME benchmark, with scores of 57.5% and 60.6% without and with subtitles, respectively. Overall, LongVILA presents a significant advancement in the realm of long-context visual-language models, providing both theoretical contributions and practical solutions to the challenges posed by long video understanding.

**Strengths:**

* The paper is well-written and easy to understand. The figure illustration and captions are informative.
* The authors provide a full-stack design for long-context Vision-Language Models, including both training curriculum and system implementation. These contributions are significant in the multimodal foundation model community.
* The proposed model, LongVILA, presents strong performance on VideoMME and other long video understanding tasks.
* The Multi-Modal Sequence Parallelism design can greatly reduce the memory cost in both training and inference.

**Weaknesses:**

* The authors do not seem to provide performance metrics on general video understanding benchmarks, such as the Perception Test [A] or EgoSchema [B], after improving long video understanding capabilities. It is worth noting whether the performance on general benchmarks is affected after enhancing long-context capabilities.
* The proposed Multi-Modal Sequence Parallelism design is interesting and effective. However, it is not clear whether the source code will be released, which will be beneficial to the community.

---

[A]Patraucean, V., Smaira, L., Gupta, A., Recasens, A., Markeeva, L., Banarse, D., ... & Carreira, J. (2024). Perception test: A diagnostic benchmark for multimodal video models. Advances in Neural Information Processing Systems, 36.

[B]Mangalam, K., Akshulakov, R., & Malik, J. (2024). Egoschema: A diagnostic benchmark for very long-form video language understanding. Advances in Neural Information Processing Systems, 36.

**Questions:**

Refer to weakness.

---

> ### Author Response · Authors · 2024-11-20
> **Response to Reviewer 7Lyg**
>
> We are truly appreciated for your valuable comments. In the following, we provide responses to the concerns.
>
> **Q1: “Performance metrics on general video understanding benchmarks, such as the PerceptionTest or EgoSchema.”**
>
> **A:** Thank you for the constructive suggestion. We have included 8 additional video benchmarks for evaluation, with ActivityNet [1], EgoSchema [2], PerceptionTest [3], MVBench [4], and NExT-QA [5] serving as general video understanding benchmarks. LongVILA-7B demonstrates promising performance on these benchmarks compared to strong baselines such as LLaVA-OneVision [6]. This is **Table 3 in the revision**.
>
> | Model | LLM Size | ActNetQA | EgoSchema | PercepTest | MVBench | NExTQA | LVideoBench | EventBench | VNBench | VideoMME | VideoMME |
> | --- | --- | --- | --- | --- | --- | --- | --- | --- | --- | --- | --- |
> |  |  | test | test | val | test | mc | val | val | val | w/o sub. | w/ sub. |
> | GPT-4V | - | 57.0 | - | - | 43.5 | - | 61.3 | 32.6 | - | 59.9 | 63.3 |
> | GPT-4o | - | - | - | - | - | - | 66.7 | 53.3 | 64.4 | 71.9 | 77.2 |
> | Gemini-1.5-Pro | - | 57.5 | 72.2 | - | - | - | 64.0 | 43.2 | 66.7 | 75.0 | 81.3 |
> |  |  |  |  |  |  |  |  |  |  |  |  |
> | Video-LLaVA | 7B | 45.3 | 38.4 | - | 43.5 | - | 37.6 | 5.9 | 12.4 | 39.9 | 41.6 |
> | Flash-VStream | 7B | 51.9 | - | - | - | 61.6 | - | - | - | - | - |
> | ShareGPT4Video | 8B | 50.8 | - | - | 51.2 | - | 41.8 | - | - | 39.9 | 43.6 |
> | VideoLLaMA2 | 7B | 50.2 | 51.7 | 51.4 | 54.6 | - | - | 6.9 | 4.5 | 47.9 | 50.3 |
> | VideoLLaMA2.1 | 7B | 53.0 | 53.1 | 54.9 | 57.3 | - | - | - | - | 54.9 | 56.4 |
> | Kangaroo | 8B | - | 62.7 | - | 61.1 | - | 54.8 | - | - | 56.0 | 57.6 |
> | PLLaVA | 7B | 56.3 | - | - | 46.6 | - | 39.2 | 28.2 | - | - | - |
> | LLaVA-OV | 7B | 56.7 | 60.1 | 57.1 | 56.7 | 79.4 | 56.4 | - | 51.8 | 58.2 | 61.5 |
> | **LongVILA** | **7B** | **59.5** | **67.7** | **58.1** | **67.1** | **80.7** | **57.1** | **58.0** | **63.0** | **60.1** | **65.1** |
>
> **Q2: “The proposed Multi-Modal Sequence Parallelism design and the source code.”**
>
> **A:** The source code, including the Multi-Modal Sequence Parallelism implementation, will definitely be released. For your convenience, we provide an [anonymous link](https://anonymous.4open.science/r/LongVILA) of our internal source code, where Multi-Modal Sequence Parallelism training can be easily enabled using the script with “llava/train/train_hybrid.py” and the argument “--seq_parallel_size”.
>
> [1] Zhou Yu, Dejing Xu, Jun Yu, Ting Yu, Zhou Zhao, Yueting Zhuang, Dacheng Tao: ActivityNet-QA: A Dataset for Understanding Complex Web Videos via Question Answering. AAAI 2019
>
> [2] Karttikeya Mangalam, Raiymbek Akshulakov, Jitendra Malik: EgoSchema: A Diagnostic Benchmark for Very Long-form Video Language Understanding. NeurIPS 2023
>
> [3] Viorica Patraucean, et. al: Perception Test: A Diagnostic Benchmark for Multimodal Video Models. NeurIPS 2023
>
> [4] Kunchang Li, Yali Wang, Yinan He, Yizhuo Li, Yi Wang, Yi Liu, Zun Wang, Jilan Xu, Guo Chen, Ping Lou, Limin Wang, Yu Qiao: MVBench: A Comprehensive Multi-modal Video Understanding Benchmark. CVPR 2024
>
> [5] Junbin Xiao, Xindi Shang, Angela Yao, Tat-Seng Chua: NExT-QA: Next Phase of Question-Answering to Explaining Temporal Actions. CVPR 2021
>
> [6] Bo Li, Yuanhan Zhang, Dong Guo, Renrui Zhang, Feng Li, Hao Zhang, Kaichen Zhang, Yanwei Li, Ziwei Liu, Chunyuan Li: LLaVA-OneVision: Easy Visual Task Transfer. 2024

---

> > ### Author Response · Authors · 2024-11-25
> > **Follow-Up on Review and Feedback**
> >
> > Dear Reviewer **7Lyg**,
> >
> > We hope this message finds you well.
> >
> > We have carefully addressed all your questions and concerns, including conducting additional experiments as requested, and have provided detailed responses in the rebuttal.
> >
> > As the rebuttal deadline is approaching, we would deeply appreciate it if you could share your updated thoughts based on the rebuttal and paper revision, or do not hesitate to let us know if you have additional questions, and we will respond promptly.
> >
> > Thank you again for your thoughtful review and your invaluable contributions to the quality of this paper.
> >
> >
> > Kind regards,
> >
> > Paper 2006 Authors

---

> > ### Comment · Reviewer_7Lyg · 2024-11-25
> >
> > Thanks to the authors for addressing my concerns. I think my questions have been sufficiently addressed and I have raised my score. I hope the authors can release the source codes for the Multi-Modal Sequence Parallelism design as promised, which is a key contribution of this paper.

---

> > > ### Author Response · Authors · 2024-11-25
> > > **Reply to Reviewer 7Lyg**
> > >
> > > Thank you again for your valuable comments and for recognizing this work. Please rest assured that we will release all our code and models associated with this work, as promised, including MM-SP.

---

### Official Review · Reviewer_DDGN · 2024-11-04

**Soundness:** 4
**Presentation:** 4
**Contribution:** 4
**Rating:** 6
**Confidence:** 4

**Summary:**

The paper proposes a novel long-context Multi-Modal Sequence Parallelism (MM-SP) system specifically designed to enhance the performance of Vision-Language Models (VLMs) when processing extended contexts, increasing the window size from 8 to 1024 frames. The authors demonstrate how their approach can effectively manage and leverage longer input sequences, aiming to address limitations in current VLMs regarding video processing capabilities.

**Strengths:**

- The introduction of MM-SP is a significant contribution to the field, as it creatively combines techniques for managing long-context sequences in a multi-modal setting. This approach addresses a gap in existing models that struggle with extensive video inputs.

**Weaknesses:**

- The evaluation primarily utilizes a single benchmark (VideoMME), which may not sufficiently demonstrate the capabilities of the proposed system in handling long videos. It would be beneficial to incorporate additional benchmarks focused on long video analysis, such as those introduced in recent works (e.g., arXiv:2406.09367 and arXiv:2406.14129) to validate the robustness of the findings.
- While the authors mention the ability to process 1024 frames, the results are only provided for 256 frames (Figure 2(a)). It would strengthen the paper to include performance metrics for 512 and 1024 frames across the discussed benchmarks to fully substantiate the claims made.

**Questions:**

- If the authors were to mix the stage 3 with stage 5 at the end, which is stage 1-2-4-(3&5), what anticipated effects on performance or learning dynamics might arise? Would the integration lead to any synergies or drawbacks?
- How would the performance change if context extension were prioritized by using a long-window LLM first, followed by stages 4-1-2-3-5?
- How would the performance change if you conbine these two stratgies, which is 4-1-2-(3&5)?

---

> ### Author Response · Authors · 2024-11-20
> **Response to Reviewer DDGN**
>
> We sincerely appreciate your valuable comments. Below, we provide responses to each concern.
>
> **Q1: “Additional benchmarks focused on long video analysis, such as those introduced in recent works.”**
>
> **A:** We have evaluated LongVILA on 8 additional benchmarks, including the two you suggested: EventBench (arXiv:2406.14129) [1] and VNBench (arXiv:2406.09367) [2], which focus on long video analysis. LongVILA-7B demonstrates strong performance across these benchmarks, achieving the best results on EventBench. On VNBench, LongVILA-7B is slightly outperformed by GPT-4o [3] and Gemini-1.5-Pro [4], but surpasses other open models. These results are presented in **Table 3 in the revision**.
>
> | Model | LLM Size | EventBench | VNBench | ActNetQA | EgoSchema | LVideoBench | PercepTest | MVBench | NExTQA | VideoMME | VideoMME |
> | --- | --- | --- | --- | --- | --- | --- | --- | --- | --- | --- | --- |
> |  |  | val | val | test | test | val | val | test | mc | w/o sub. | w/ sub. |
> | GPT-4V | - | 32.6 | - | 57.0 | - | 61.3 | - | 43.5 | - | 59.9 | 63.3 |
> | GPT-4o | - | 53.3 | 64.4 | - | - | 66.7 | - | - | - | 71.9 | 77.2 |
> | Gemini-1.5-Pro | - | 43.2 | 66.7 | 57.5 | 72.2 | 64.0 | - | - | - | 75.0 | 81.3 |
> |  |  |  |  |  |  |  |  |  |  |  |  |
> | Video-LLaVA | 7B | 5.9 | 12.4 | 45.3 | 38.4 | 37.6 | - | 43.5 | - | 39.9 | 41.6 |
> | Flash-VStream | 7B | - | - | 51.9 | - | - | - | - | 61.6 | - | - |
> | ShareGPT4Video | 8B | - | - | 50.8 | - | 41.8 | - | 51.2 | - | 39.9 | 43.6 |
> | VideoLLaMA2 | 7B | 6.9 | 4.5 | 50.2 | 51.7 | - | 51.4 | 54.6 | - | 47.9 | 50.3 |
> | VideoLLaMA2.1 | 7B | - | - | 53.0 | 53.1 | - | 54.9 | 57.3 | - | 54.9 | 56.4 |
> | Kangaroo | 8B | - | - | - | 62.7 | 54.8 | - | 61.1 | - | 56.0 | 57.6 |
> | PLLaVA | 7B | 28.2 | - | 56.3 | - | 39.2 | - | 46.6 | - | - | - |
> | LLaVA-OV | 7B | - | 51.8 | 56.7 | 60.1 | 56.4 | 57.1 | 56.7 | 79.4 | 58.2 | 61.5 |
> | LongVILA | 7B | **58.0** | **63.0** | **59.5** | **67.7** | **57.1** | **58.1** | **67.1** | **80.7** | **60.1** | **65.1** |
>
> **Q2: “To include performance metrics for 512 and 1024 frames in Figure 2(a).”**
>
> **A:** Thank you for the suggestion. We have revised the figure (now **Figure 3**) to include performance metrics for 512 and 1024 frames, showing sustained growth as the number of frames increases.
>
> **Q3: “The performance change of stages 1-2-4-(3&5), 4-1-2-3-5, 4-1-2-(3&5).”**
>
> **A:** Thank you for this insightful question. We conducted additional ablation studies on these training stage configurations, using VideoMME without subtitles for evaluation. Mixing stage 3 with stage 5 (i.e., **1-2-4-(3&5)**) leads to large performance drops on short and medium videos. When context extension is prioritized (i.e., **4-1-2-3-5**), performance degrades for long videos, likely due to the extensive use of short data in stages 2 and 3, which diminishes long context capabilities. Further mixing stage 3 with stage 5 (**4-1-2-(3&5**) again results in performance drops for short and medium videos. These results are provided in **Table 1** **in the revision.**
>
> In addition to performance, the primary reason for selecting **1-2-3-4-5** is that **1-2-3** represents a widely adopted VLM training pipeline. Leveraging pre-trained, off-the-shelf VLMs provides two key advantages:
>
> 1. It avoids the costly re-training of stages 2 (VLM pretraining) and 3 (VLM supervised fine-tuning).
> 2. Stages 1, 2, and 3 are often difficult to train for public community, as open VLMs do not always provide access to their datasets (e.g., Qwen2-VL, LLaMA3.2-VL). Fixing **1-2-3** ensures greater consistency and reliability for long-term research, which is crucial for AI conferences.
>
> | Training Stages | Average | Short | Medium | Long |
> | --- | --- | --- | --- | --- |
> | 1-2-3-4-5 | 57.5 | 69.3 | 56.1 | 47.0 |
> | 1-2-4-(3&5) | 55.9 | 67.4 | 54.1 | 46.1 |
> | 4-1-2-3-5 | 56.0 | 69.2 | 54.1 | 44.5 |
> | 4-1-2-(3&5) | 55.3 | 67.2 | 53.6 | 45.1 |
>
> [1] Yifan Du, Kun Zhou, Yuqi Huo, Yifan Li, Wayne Xin Zhao, Haoyu Lu, Zijia Zhao, Bingning Wang, Weipeng Chen, Ji-Rong Wen: Towards Event-oriented Long Video Understanding. 2024
>
> [2] Zijia Zhao, Haoyu Lu, Yuqi Huo, Yifan Du, Tongtian Yue, Longteng Guo, Bingning Wang, Weipeng Chen, Jing Liu: Needle In A Video Haystack: A Scalable Synthetic Framework for Benchmarking Video MLLMs. 2024
>
> [3] OpenAI. Hello gpt-4o. 2024
>
> [4] Gemini Team. Gemini 1.5: Unlocking multimodal understanding across millions of tokens of context, 2024.

---

> > ### Author Response · Authors · 2024-11-25
> > **Follow-Up on Review and Feedback**
> >
> > Dear Reviewer **DDGN**,
> >
> > We hope this message finds you well.
> >
> > We have carefully addressed all your questions and concerns, including conducting additional experiments as requested, and have provided detailed responses in the rebuttal.
> >
> > As the rebuttal deadline is approaching, we would deeply appreciate it if you could share your updated thoughts based on the rebuttal and paper revision, or do not hesitate to let us know if you have additional questions, and we will respond promptly.
> >
> > Thank you again for your thoughtful review and your invaluable contributions to the quality of this paper.
> >
> >
> > Kind regards,
> >
> > Paper 2006 Authors

---

> > > ### Comment · Reviewer_DDGN · 2024-11-26
> > >
> > > Thank you for your response. My concerns have been addressed. I will keep my score at the current rating.

---

> > > > ### Author Response · Authors · 2024-11-26
> > > > **Reply to Reviewer DDGN**
> > > >
> > > > Thanks again for your invaluable feedback and insightful comments, which have profoundly enriched and elevated the quality of this work.

---

### Author Response · Authors · 2024-11-20
**General Response**

We appreciate all reviewers for their detailed and constructive feedback. We have revised the paper accordingly. We upload the revised paper, with major updates highlighted in blue.

1. **Expanded Evaluation**: We have included **8 additional video benchmarks** for evaluation: ActivityNet-QA [1], EgoSchema [2], EventBench [3], LongVideoBench [4], PerceptionTest [5], MVBench [6], NExT-QA [7], and VNBench [8]. Alongside VideoMME [9], we now compare LongVILA against state-of-the-art methods on a total of **9 benchmarks**, demonstrating consistently strong performance. The results are shown in **Table 3 in the revision**.
2. **Needle in the Long Video Haystack Experiment**: We report stronger results, with the LongVILA model trained on 2048 frames achieving 99.8% accuracy on **6,000 frames (exceeding 1 million tokens)**. The results are illustrated in **Figure 2 in the revision**.
3. **VideoMME Update**: We have updated our results on **VideoMME** by adding the LongVILA-1.5B model and results for 256 frames. LongVILA-7B achieves **60.1% / 65.1%** accuracy for the settings without and with subtitles, respectively, as shown in **Table 4 in the revision**.
4. **Model Complexity Analysis**: We present a detailed analysis of **model complexity** across various factors including model size, components, number of frames, context length, latency, and FLOPs in **Table 10 in the revision**.
5. **Additional Baselines**: We compare our model against more baselines, including proprietary models (GPT-4V [10], GPT-4o [11], Gemini-1.5-Pro [12]) and open models (Flash-VStream [13], VideoLLaMA2.1 [14], PLLaVA [15], LLaVA One-Vision [16]), as presented in **Table 3 in the revision**.
6. **Additional Ablations**: We conducted further ablations on training schedule settings, as presented in **Table 1 in the revision**.

The following sections provide detailed responses to all reviewer comments.

[1] Zhou Yu, Dejing Xu, Jun Yu, Ting Yu, Zhou Zhao, Yueting Zhuang, Dacheng Tao: ActivityNet-QA: A Dataset for Understanding Complex Web Videos via Question Answering. AAAI 2019

[2] Karttikeya Mangalam, Raiymbek Akshulakov, Jitendra Malik: EgoSchema: A Diagnostic Benchmark for Very Long-form Video Language Understanding. NeurIPS 2023

[3] Yifan Du, Kun Zhou, Yuqi Huo, Yifan Li, Wayne Xin Zhao, Haoyu Lu, Zijia Zhao, Bingning Wang, Weipeng Chen, Ji-Rong Wen: Towards Event-oriented Long Video Understanding. 2024

[4] Haoning Wu, Dongxu Li, Bei Chen, Junnan Li: LongVideoBench: A Benchmark for Long-context Interleaved Video-Language Understanding. 2024

[5] Viorica Patraucean, et. al: Perception Test: A Diagnostic Benchmark for Multimodal Video Models. NeurIPS 2023

[6] Kunchang Li, Yali Wang, Yinan He, Yizhuo Li, Yi Wang, Yi Liu, Zun Wang, Jilan Xu, Guo Chen, Ping Lou, Limin Wang, Yu Qiao: MVBench: A Comprehensive Multi-modal Video Understanding Benchmark. CVPR 2024

[7] Junbin Xiao, Xindi Shang, Angela Yao, Tat-Seng Chua: NExT-QA: Next Phase of Question-Answering to Explaining Temporal Actions. CVPR 2021

[8] Zijia Zhao, Haoyu Lu, Yuqi Huo, Yifan Du, Tongtian Yue, Longteng Guo, Bingning Wang, Weipeng Chen, Jing Liu: Needle In A Video Haystack: A Scalable Synthetic Framework for Benchmarking Video MLLMs. 2024

[9] Chaoyou Fu, e t. al.: Video-MME: The First-Ever Comprehensive Evaluation Benchmark of Multi-modal LLMs in Video Analysis. 2024

[10] OpenAI. Gpt-4v. 2023

[11] OpenAI. Hello gpt-4o. 2024

[12] Gemini Team. Gemini 1.5: Unlocking multimodal understanding across millions of tokens of context, 2024.

[13] Haoji Zhang, Yiqin Wang, Yansong Tang, Yong Liu, Jiashi Feng, Jifeng Dai, Xiaojie Jin: Flash-VStream: Memory-Based Real-Time Understanding for Long Video Streams. 2024

[14] Zesen Cheng, et. al.: VideoLLaMA 2: Advancing Spatial-Temporal Modeling and Audio Understanding in Video-LLMs. 2024

[15] Lin Xu, Yilin Zhao, Daquan Zhou, Zhijie Lin, See-Kiong Ng, Jiashi Feng: PLLaVA : Parameter-free LLaVA Extension from Images to Videos for Video Dense Captioning. 2024

[16] Bo Li, Yuanhan Zhang, Dong Guo, Renrui Zhang, Feng Li, Hao Zhang, Kaichen Zhang, Yanwei Li, Ziwei Liu, Chunyuan Li: LLaVA-OneVision: Easy Visual Task Transfer. 2024

---

### Meta-Review · Area_Chair_GLx7 · 2024-12-18

**Metareview:**

The paper proposes a novel long-context Multi-Modal Sequence Parallelism system to enhance Vision-Language Models in processing extended contexts, increasing the VILA model's capability from 8 to 1024 frames. Extensive experiments conducted after the rebuttal stage demonstrate the effectiveness of their approach in managing and leveraging longer input sequences.

This paper received consistent positive scores, and the ACs recommend it for acceptance as a spotlight paper.

**Additional Comments On Reviewer Discussion:**

Reviewer DDGN considers the introduction of MM-SP a significant contribution to the field, as it effectively manages long-context sequences in a multi-modal setting. The reviewer notes the insufficient experimentation on a single benchmark, which was addressed in the rebuttal.

Reviewers 7Lyg and MTsu also pointed out the lack of experiments on other general video understanding benchmarks. The rebuttal provided additional experiments to address this concern.

After the rebuttal, the authors demonstrated the effectiveness of their approach with more comprehensive experiments.

---

### Decision · Program_Chairs · 2025-01-22

Accept (Poster)